# INSTRUCTPLM-MU: 1-HOUR FINE-TUNING OF ESM2 BEATS ESM3 IN PROTEIN MUTATION PREDICTIONS

## ABSTRACT

Multimodal protein language models deliver strong performance on mutation-effect prediction, but training such models from scratch demands substantial computational resources. In this paper, we propose a fine-tuning framework called InstructPLM-mu and try to answer a question: *Can multimodal fine-tuning of a pretrained, sequence-only protein language model match the performance of models trained end-to-end?* Surprisingly, our experiments show that fine-tuning ESM2 with structural inputs can reach performance comparable to ESM3. To understand how this is achieved, we systematically compare three different feature-fusion designs and fine-tuning recipes. Our results reveal that both the fusion method and the tuning strategy strongly affect final accuracy, indicating that the fine-tuning process is not trivial. We hope this work offers practical guidance for injecting structure into pretrained protein language models and motivates further research on better fusion mechanisms and fine-tuning protocols.

## 1 INTRODUCTION

Proteins are vital macromolecules that perform a diverse array of cellular functions, from catalyzing biochemical reactions to maintaining structural integrity and regulating signaling pathways. These functions are determined by the protein's three-dimensional structure, which in turn is encoded by its amino acid sequence (Bertoline et al., 2023; Kim et al., 2025). During natural evolution, mutations inevitably arise in protein sequences. While most are random, their long-term persistence is shaped by selective pressures that favor variants better adapted to their environments (Hie et al., 2024). Changes at specific residues can significantly impact a protein's folding stability, functional fitness, or biochemical activity (Parthiban et al., 2006; Wang et al., 2020; Boehr et al., 2009; Sonaglioni et al., 2024; Albanese et al., 2025). While it sometimes leads to a complete loss of function or even toxic effects. Such mutational outcomes are central to both the emergence of new protein functions and the molecular basis of genetic diseases.

Deep mutational scanning (DMS) is an experimental technique that systematically measures the functional impact of a vast number of sequence variants for a given protein (Fowler et al., 2014; Fowler & Fields, 2014; Hanning et al., 2022). By introducing and testing millions of mutations, DMS generates high-resolution maps that link sequence changes to functional outcomes. These datasets have become invaluable for understanding sequence-function relationships, guiding protein engineering, and benchmarking computational prediction models. However, due to the high cost and limited throughput of DMS experiments, it is infeasible to apply them broadly across all proteins or mutation types (Fowler & Fields, 2014). To address this limitation, numerous computational methods have been proposed to predict mutational effects based on sequence features (Lin et al., 2023; Marquet et al., 2024), structural features (Su et al., 2023; Zhang et al., 2024; Sun et al., 2025), and evolutionary information (Meier et al., 2021; Weitzman et al., 2025; Tan et al., 2025a; Sun et al., 2024). Among these, multimodal protein language models (PLMs) have demonstrated strong generalization capabilities, leveraging large-scale unlabeled protein sequences to capture evolutionary and biochemical constraints without explicit supervision (Notin et al., 2023).

However, training multimodal protein language models from scratch typically demands substantial computational resources and large-scale annotated datasets, making such approaches impractical for many researchers (Su et al., 2024). Inspired by recent advances in vision-language models, several studies have explored the use of pretrained PLMs as a backbone, and then fine-tuning them with

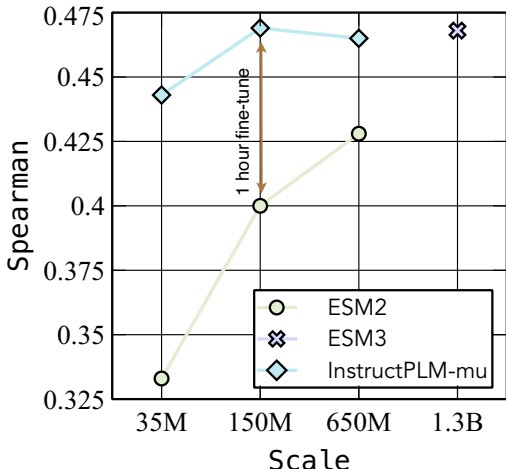

Figure 1: Protein mutation prediction performance of InstructPLM-mu and ESM3. After 1 hour of fine-tuning, InstructPLM-mu on the 150M ESM2 backbone overtakes ESM3.

additional modalities (e.g., structure or evolutionary data) (Zheng et al., 2023; Qiu et al., 2024; Ruffolo et al., 2024). These approaches have demonstrated promising results while remaining data- and resource-efficient. Nevertheless, how to effectively integrate new modalities into pretrained models remains an open research question, with design choices in fusion strategies playing a critical role in downstream performance.

In this paper, we aim to find an efficient way to fuse structure embeddings into protein language models, attempting to answer the question: *Can multimodal fine-tuning achieve comparable or even surpass the multimodal model trained from scratch?* Specifically, we propose a multimodal fine-tuning framework called InstructPLM-mu, and investigate three different strategies: Cross Attention, Channel-wise Concat, and Token-wise Concat. We apply our methods to the widely used protein language model ESM2 (Lin et al., 2023) and two representative structure encoders, Protein-MPNN (Dauparas et al., 2022) and ESM-IF (Hsu et al., 2022), to evaluate the performance on the zero-shot protein mutation prediction task. Through extensive experiments, our results show that fine-tuned models can match the performance of advanced multimodal methods, even surpassing the ESM3 (Hayes et al., 2025), which is a newer, bigger, and multimodal successor of ESM2. More importantly, our ablation shows that choosing the fine-tuning strategy is also critical; an overly aggressive training recipe may lead to knowledge forgetting of the pretrained protein language models. We will release code and checkpoints to facilitate reproducibility.

## 2 RELATED WORKS

### 2.1 PROTEIN MUTATION PREDICTION

Protein mutation prediction is central to understanding protein function and guiding protein engineering. Classical approaches such as Rosetta(Alford et al., 2017) and ABACUS2(Xiong et al., 2020) rely on energy-based scoring, but are hindered by sampling limitations and biases in their underlying potentials. Deep learning has opened new directions: models like ESM-1v(Meier et al., 2021) predict mutation effects from large-scale sequence data, while structure-aware methods such as ProSST(Li et al., 2024b) and Pythia(Sun et al., 2025) further improve accuracy by incorporating structural information.

Benchmarking efforts such as ProteinGym (Notin et al., 2023) have underscored the power of pre-trained PLMs in protein modeling. ProteinGym evaluated over 250 deep mutational scanning assays, revealing that PLMs effectively capture evolutionary constraints and generalize across diverse proteins. For instance, AIDO.Protein (Sun et al., 2024), a state-of-the-art PLM with a mixture-of-experts architecture, highlights the potential of PLMs to scale up protein modeling tasks with enhanced computational efficiency. Meanwhile, models like VenusREM (Tan et al., 2024) and

S3F (Zhang et al., 2024) leverage multimodal information—integrating structure, sequence, evolution, and surface features—to precisely model local conformational and energetic changes, which significantly boosts model performance.

## 2.2 Multimodal Alignment

Multimodal alignment aligns heterogeneous features across modalities to enhance cross-modal understanding and specific task performance (Baltrušaitis et al., 2018). This field has gained prominence with advances in large language and vision models (Alayrac et al., 2022). Key challenges in multimodal learning include Feature Fusion and Training Paradigm, which are critical for model performance (Tong et al., 2024; Li & Tang, 2024).

**Feature Fusion.** Previous studies on multimodal alignment largely focus on the way of projection between different modalities. For example, MLP projection methods successfully bridge visual features to LLM token spaces (Liu et al., 2023; 2024; Li et al., 2024a). Query-based resampling optimizes computational efficiency through cross-attention compression of visual tokens (Bai et al., 2023). Architectures with gated or sparse cross-attention layers for deeper multimodal integration (Alayrac et al., 2022; Awadalla et al., 2023). The main goal of these methods is to discuss how to deal with images with different resolutions and scales. However, recent work highlights that not only the manipulation of multimodal features but also the manner in which these features are fused inside the language model is crucial. DeepStack (Meng et al., 2024), for instance, demonstrates that multimodal performance can be enhanced by injecting vision features into multiple layers of the LLM. Similarly, a recent study systematically examines four different fusion strategies across a broad range of NLP tasks (Lin et al., 2025). Despite these advances, multimodal fusion strategies remain underexplored in the context of protein language models. Notably, unlike vision–language models, protein structural features can be naturally aggregated at the residue level, which facilitates fine-grained integration of structural signals into sequence representations and opens up opportunities for designing more efficient and biologically informed fusion mechanisms.

**Training Paradigm.** The end-to-end training paradigm jointly optimizes all parameters in a single phase, pursuing global optimization at the cost of high computational demand and potential suboptimal alignment due to limited intermediate refinement (Tong et al., 2024). In contrast, multi-stage training separately fine-tunes modality-specific modules (e.g., image encoder) before full-model optimization, improving efficiency and final performance (Liu et al., 2023; Wadekar et al., 2024; Wu et al., 2025). Unified pretraining integrates multimodal inputs within a single framework, typically employing masked or autoregressive objectives to achieve cross-modal fusion (Zhu et al., 2025). Additionally, strategies such as reinforcement learning have been proposed for supervision-efficient alignment in specific contexts (Sun et al., 2023; Chu et al., 2025).

Inspired by advances in vision-language multimodal alignment, similar sequence-structure alignment strategies are now being adapted for protein modeling (Su et al., 2023; Li et al., 2024b; Qiu et al., 2024; Hayes et al., 2025). Here, we compare multiple alignment mechanisms between pre-trained sequence and structure modules, evaluating their performance on protein mutation prediction (Notin et al., 2023). This task is central to protein engineering, as accurately predicting the functional effects of mutations (e.g., on stability, binding, and activity) requires high-quantity representation and deep integration of both sequence and structural information (Meier et al., 2021). It thus provides a rigorous test for assessing protein multimodal model quality and their ability to integrate sequence-structure relationships for precise functional inference.

## 3 Methods

In this section, we compare three different multimodal fusion strategies and highlight their key differences. We denote the embedding of the amino acid sequence with length $L$ as

$$\mathbf{X}^{(seq)} = [x_1, x_2, \ldots, x_L], \quad x_i \in \mathbb{R}^{d_s}, \tag{1}$$

where $d_s$ is the embedding dimension for sequence tokens after the embedding layer of PLMs. The corresponding protein structure encoder produces a structural embedding

$$\mathbf{X}^{(str)} = [s_1, s_2, \ldots, s_L], \quad s_i \in \mathbb{R}^{d_t}, \tag{2}$$

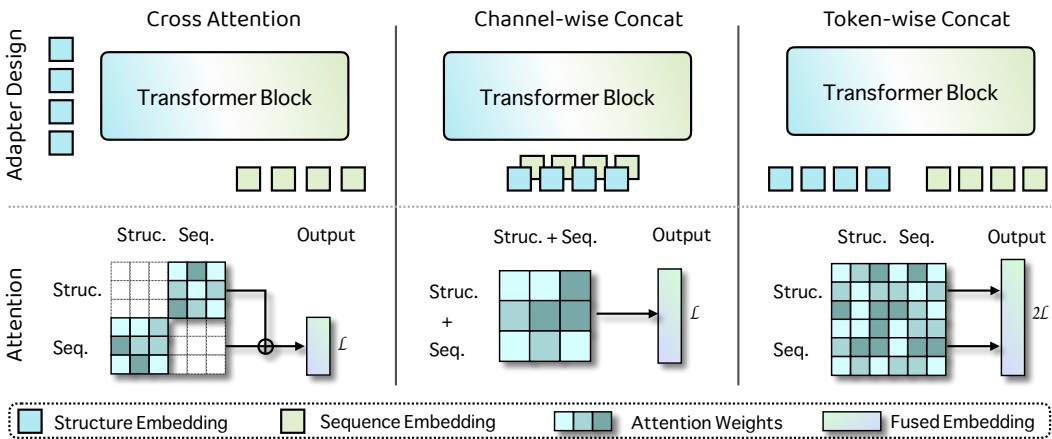

Figure 2: Comparison of three different multimodal fusion strategies of InstructPLM-mu: cross attention (Left), Channel-wise Concat (Middle), and Token-wise Concat (Right).

where $d_t$ is the structural embedding dimension. Before the structure embeddings are fused to sequence embeddings, we use a learned multi-layer perceptron (MLP) $\mathbf{W}_{str} \in \mathbb{R}^{d_t \times d_s}$ to match the dimensions between modalities (Liu et al., 2023):

$$\tilde{\mathbf{X}}^{(str)} = \mathrm{MLP}(\mathbf{X}^{(str)}), \quad \tilde{\mathbf{X}}^{(str)} \in \mathbb{R}^{L \times d_s}. \tag{3}$$

### 3.1 CROSS ATTENTION

In the Cross Attention strategy, we follow the implementation of LM-Design (Zheng et al., 2023), insert an additional Cross Attention sub-layer into the final transformer block of the PLM. (Fig. 2 left.) Let $\mathbf{H}^{(L-1)} \in \mathbb{R}^{L \times d_s}$ be the sequence representation output from the $(L-1)$-th Transformer block (i.e., before the final block). The final block is modified to include an additional Cross Attention branch:

$$\mathbf{H}^{(seq)}_{\mathrm{self}} = \mathrm{SelfAttn}(\mathbf{H}^{(L-1)}) \tag{4}$$

$$\mathbf{H}^{(seq)}_{\mathrm{cross}} = \mathrm{CrossAttn}\left(Q = \mathbf{H}^{(seq)}_{\mathrm{self}}, \; K = \tilde{\mathbf{X}}^{(str)}, \; V = \tilde{\mathbf{X}}^{(str)}\right) \tag{5}$$

$$\mathbf{H}^{(L)} = \mathbf{H}^{(seq)}_{\mathrm{self}} + \mathbf{H}^{(seq)}_{\mathrm{cross}} \tag{6}$$

The fused representation $\mathbf{H}^{(L)}$ is then passed to the PLM's output layer for prediction.

The Cross Attention design has several limitations. First, since the structure embeddings $\tilde{\mathbf{H}}^{(str)}$ serve only as the key and value in the Cross Attention operation, there is *no self-attention mechanism among structural tokens themselves*. This prevents the structure modality from refining its internal representation or capturing long-range dependencies purely within the structural space during fusion. Second, this one-way attention flow (sequence $\rightarrow$ structure) does not allow reciprocal updates from structure to sequence across layers; consequently, the structure features will not merge information from sequences.

### 3.2 CHANNEL-WISE CONCAT

In the Channel-wise Concat strategy, structural features are directly merged with sequence features at the embedding level; this strategy is adopted by ESM 3 (Hayes et al., 2025). (Fig. 2 middle.) Specifically, the projected structure embeddings are added element-wise to the sequence embeddings:

$$\mathbf{Z} = \mathbf{X}^{(seq)} + \tilde{\mathbf{X}}^{(str)}, \quad \mathbf{Z} \in \mathbb{R}^{L \times d_s}. \tag{7}$$

The fused representation $\mathbf{Z}$ is fed into the PLM in place of the original sequence embeddings.

Compared to Cross Attention, Channel-wise concatenation enables attention on both structure and sequences: when the PLM performs self-attention over $\mathbf{Z}$, information from the structure and sequence modalities can flow jointly. However, the tight coupling also means there is no mechanism

Table 1: Summary of datasets used in InstructPLM-mu. The training and validation sets are derived from CATH 4.3, while evaluation is performed on the ProteinGYM benchmarks, covering activity, binding, expression, fitness, and stability. The table lists the number of sequences or mutational assays in each split.

| Dataset | CATH 4.3 | | ProteinGYM | | | | |
|---|---|---|---|---|---|---|---|
| | train | validation | Acitivity | Binding | Expression | Fitness | Stability |
| Number | 22727 | 2525 | 39 | 12 | 16 | 69 | 66 |

for *selective* information flow, structural features cannot be dynamically weighted or ignored depending on context. In other words, the model treats the sum of sequence and structure features as a single representation, which may lead to suboptimal integration when one modality contains noisy or task-irrelevant information.

### 3.3 TOKEN-WISE CONCAT

In the Token-wise Concat strategy, structural embeddings are treated as additional input tokens, enabling the PLM to process sequence and structure jointly through its self-attention mechanism. (Fig. 2 right.) Given the projected structure embedding $\tilde{\mathbf{X}}^{(str)}$, we concatenate the structural tokens and the sequence tokens along the sequence dimension:

$$\mathbf{Z} = [\tilde{s}_1, \ldots, \tilde{s}_L, x_1, \ldots, x_L] \in \mathbb{R}^{2L \times d_s}. \tag{8}$$

To ensure alignment between modalities, we assign the same position index to $\tilde{s}_i$ and $x_i$, so that the $i$-th structural token corresponds to the $i$-th amino acid in the sequence.

We refer to this design as InstructPLM-mu, highlighting its ability to inject *instruction-like* structural tokens into the PLM. Unlike channel-wise fusion, which enforces a static combination of modalities, InstructPLM-mu enables dynamic, bidirectional information flow between sequence and structure tokens through self-attention. This design allows the model to flexibly attend to or ignore structural cues depending on context, thereby offering richer cross-modal interactions. However, doubling the sequence length increases computational cost and memory usage, particularly for large $L$.

### 3.4 TRAINING TARGET

For all three fusion strategies, we adopt a masked language modeling (MLM) objective applied to the protein sequence. Given a wild-type amino acid sequence $\mathbf{X}^{(seq)}$, we randomly select a subset of positions $\mathcal{M} \subset \{1, \ldots, L\}$ to mask. The corresponding residues are replaced with a special [MASK] token in the sequence branch (align with the original protein language model), while the structural tokens remain unchanged. The model is trained to recover the original amino acid at each masked position:

$$\mathcal{L}_{\text{MLM}} = -\sum_{i \in \mathcal{M}} \log p_\theta\big(x_i \,\big|\, \mathbf{X}^{(seq)}_{\backslash \mathcal{M}}, \mathbf{X}^{(str)}\big) \tag{9}$$

where $p_\theta$ denotes the output distribution of the PLM with fused modalities. This setup forces the model to leverage structural context to reconstruct masked residues, encouraging it to learn complementary relationships between sequence and structure.

### 3.5 ZERO-SHOT PROTEIN MUTATION PREDICTION

We use masked-marginals (Meier et al., 2021) to calculate the mutation score. Let $\mathbf{X}^{(seq,wt)}$ be the wild-type sequence and $\mathbf{X}^{(seq,mut)}$ a mutant; let $\mathcal{M}$ be the set of mutated positions. For each $i \in \mathcal{M}$ we form a masked input by replacing the residue at $i$ with [MASK] in the sequence branch while keeping structural tokens unchanged. The per-site score is

$$s_i \;=\; \log p_\theta\big(x_i^{(mut)} \,\big|\, \mathbf{X}^{(seq,mut)}_{\backslash i}, \mathbf{X}^{(str)}\big) \;-\; \log p_\theta\big(x_i^{(wt)} \,\big|\, \mathbf{X}^{(seq,mut)}_{\backslash i}, \mathbf{X}^{(str)}\big). \tag{10}$$

The final mutant score is the sum over mutated sites:

$$S(\mathbf{X}^{(seq,mut)}) \;=\; \sum_{i \in \mathcal{M}} s_i. \tag{11}$$

Table 2: Comparison of multimodal fine-tuning strategies against single-modal PLMs on zero-shot mutation effect prediction. The reported values are Spearman correlation coefficients; higher values indicate better predictive performance, best and second best results are shown in **bold** and underlines respectively.

| Method | Average | Activity | Binding | Expression | Fitness | Stability |
|---|---|---|---|---|---|---|
| ESM2 (35M) | 0.333 | 0.325 | 0.32 | 0.357 | 0.224 | 0.437 |
| ESM2 (150M) | 0.4 | 0.405 | 0.358 | 0.422 | 0.308 | 0.507 |
| ESM2 (650M) | 0.428 | 0.44 | 0.369 | 0.44 | 0.372 | 0.52 |
| ESM2 (3B) | 0.421 | 0.434 | 0.351 | 0.429 | 0.382 | 0.507 |
| Channel-wise concat | 0.435 | 0.438 | 0.394 | 0.443 | 0.378 | 0.524 |
| Cross attention | 0.44 | 0.453 | 0.329 | 0.428 | **0.394** | 0.597 |
| Token-wise concat | **0.469** | **0.462** | **0.414** | **0.466** | 0.389 | **0.614** |
| Relative Gain | 9.5% | 5.0% | 12.2% | 5.9% | 3.1% | 18.1% |

Higher $S$ indicates the model favors the mutant residues over the wild type under the given structural context.

## 4 EXPERIMENTS

### 4.1 IMPLEMENTATION DETAILS

We train InstructPLM-mu using the CATH 4.3 dataset (Sillitoe et al., 2021) and tested on the ProteinGYM benchmark (Notin et al., 2023). We randomly split the CATH 4.3 dataset into train and validation with a ratio of 9:1, and perform checkpoint selection using the validation loss. To accelerate the fine-tuning process, we crop the training sequence to a maximum of 512 tokens, as the token-wise concat strategy increases training cost. Notably, as our downstream task does not involve testing on CATH 4.3, we thus do not adopt a test split of CATH 4.3. We evidence the model's ability by examining the prediction scores of the model for mutation outcomes ( 11) and experimental scores. The original ProteinGYM benchmark comprises 217 assays. Because most baseline models can process protein sequences only up to 1,024 residues, we excluded proteins longer than 1,000 residues, resulting in a final set of 201 assays. Details of the datasets are provided in Table 1. All experiments are done with 4 Nvidia A100 GPUs; other details, including metrics, hyperparameters, can be found in the Appendix.

### 4.2 MAIN RESULTS

**Token-wise concatenation emerges as the superior fusion strategy.** In order to understand how fusion strategies influence the performance of multimodal fine-tuning, we first investigate the performance across three different structure fusion strategies. Table 2 summarizes the results of zero-shot mutation effect prediction. Among three fusion strategies, Token-wise Concat obtains the highest average score (0.469). It also achieves the best performance in four out of five functional categories, showing that this design not only improves the overall average but also delivers stable gains across different evaluation aspects. This suggests that treating structural embeddings as extra tokens allows the model to use them more flexibly, depending on the context. Notably, although Channel-wise concat and Cross Attention reach similar average scores (0.435 vs. 0.440), their strengths appear in different places. For example, cross attention performs strong correlation on *activity* and even achieves a higher score in *fitness* of Token-wise Concat, while Channel-wise Concat gives stronger results on *binding* and *expression*. This divergence suggests that the way structural features are integrated can have very different effects depending on the functional property being predicted. Overall, these results highlight the importance of fusion design: while simply incorporating structure boosts performance, enabling flexible, token-level integration of modalities yields the most consistent and robust improvements across diverse protein mutation prediction tasks.

**InstructPLM-mu consistently outperforms single-modal sequence models.** When comparing multimodal approaches against the sequence-only baselines, we observe clear and consistent improvements. For average performance, all three fusion strategies surpass ESM2 models of different

Table 3: Performance comparison of other multimodal methods. Results of InstructPLM-mu are shown in  gray . The reported values are Spearman correlation coefficients; higher values indicate better predictive performance, best and second best results are shown in **bold** and underlines respectively.

| Method | Average | Activity | Binding | Expression | Fitness | Stability |
|---|---|---|---|---|---|---|
| MULAN (Frolova et al., 2025) | 0.323 | 0.298 | 0.344 | 0.376 | 0.232 | 0.366 |
| MIF (Yang et al., 2023) | 0.382 | 0.337 | 0.320 | 0.420 | 0.310 | 0.521 |
| MIF-ST (Yang et al., 2023) | 0.400 | 0.409 | 0.306 | 0.428 | 0.375 | 0.483 |
| Channel-wise Concat | 0.435 | 0.438 | 0.394 | 0.443 | 0.378 | 0.524 |
| ESM-IF (Hsu et al., 2022) | 0.440 | 0.418 | 0.388 | 0.437 | 0.333 | 0.623 |
| Cross Attention | 0.440 | 0.453 | 0.329 | 0.428 | 0.394 | 0.597 |
| ProtSSN (Tan et al., 2025b) | 0.453 | 0.475 | 0.373 | 0.452 | 0.399 | 0.566 |
| S2F (Zhang et al., 2024) | 0.460 | 0.474 | 0.394 | 0.462 | 0.403 | 0.566 |
| SaProt (Su et al., 2023) | 0.462 | 0.477 | 0.380 | 0.486 | 0.377 | 0.591 |
| ESM3 (Hayes et al., 2025) | 0.468 | 0.449 | 0.395 | 0.466 | 0.391 | 0.640 |
| Token-wise Concat | 0.469 | 0.462 | 0.414 | 0.466 | 0.389 | 0.614 |
| S3F (Zhang et al., 2024) | 0.473 | **0.483** | 0.403 | 0.474 | 0.413 | 0.592 |
| ProSST (Li et al., 2024b) | **0.506** | 0.479 | **0.435** | **0.521** | **0.441** | **0.651** |

scales, showing that incorporating structure is more effective than simply enlarging the backbone. Specifically, our methods achieved 18.1% improvement in the stability function (from 0.52 to 0.614). In fact, the best and second-best results in every functional category are achieved by fine-tuned multimodal models, underscoring the strong advantage of leveraging structure in this setting. These results demonstrate that structural context is a key driver of performance in mutation effect prediction, and incorporating it through fine-tuning offers a more reliable path forward than scaling sequence-only PLMs.

**Multimodal fine-tuning achieves competitive results with only a fraction of the resources.** To understand whether InstructPLM-mu can catch multimodal methods that are trained from scratch, we make a comparison of the current SOTA structure-sequence methods, including SaProt (Su et al., 2023), ESM3 (Hayes et al., 2025), S3F (Zhang et al., 2024), and ProSST (Li et al., 2024b). As Table 3 shows, while methods trained from scratch obtain strong performance, they typically require an extensive training cost. For example, ProSST is trained on 8*A100 GPUs for a month (Li et al., 2024b). In contrast, our fine-tuned models build on existing models and require efficient computational capacity, not only to close the gap but also to surpass larger, scratch-trained baselines. Notably, Token-wise concat (0.469) outperforms ESM3 (0.468), despite ESM3 being trained with far greater resources. S3F achieves better performance than InstructPLM-mu by incorporating extra surface modality into the count; this does not contradict our methods, yet strengthens our conclusion that multimodal fine-tuning can significantly improve performance. Another interesting observation is that our methods consistently boost the standalone performance of the original structure encoder ESM-IF on 4 out of 5 different functions, showing that structural and sequence features reinforce each other rather than being passively combined. In summarize, these results demonstrate that fine-tuning with modality fusion offers a resource-efficient yet highly effective alternative to training multimodal PLMs from scratch.

### 4.3 Fine-tuning strategies

To teach the pretrained PLMs to understand structures, a new adapter has been introduced to connect two different modalities (Eq. 3). This raises the question of how to train the added parameters: Do different training recipes influence the final performance? To answer it, we evaluate three fine-tuning strategies: Full Fine-tune, LoRA + Adapters, and Adapter-only. These strategies differ in the fraction of tunable parameters and thus reflect different degrees of intervention on the pretrained model. Specifically, Adapter-only updates only the adapter parameters ( 1% of total parameters), directly controlling how structural embeddings are injected without modifying the backbone. LoRA + Adapters additionally applies low-rank updates to selected PLM layers ( 5–10% of total parameters), offering a middle ground between efficiency and capacity. Full Fine-tune updates all parameters of the PLM and adapters, yielding maximum flexibility at the highest computational cost.

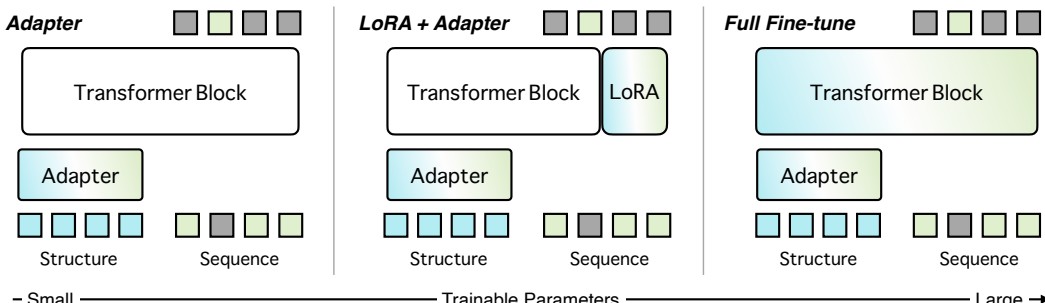

Figure 3: Schematic of the three fine-tuning strategies. **Left**, Adapter-only: the backbone is frozen and only the adapters that project structural features into the PLM are learned. **Middle**, LoRA + Adapter: adapters inject structural embeddings while low-rank (LoRA) updates are applied to selected transformer weights. **Right**, Full Fine-tune: all transformer blocks and adapter modules are updated.

Table 4: Comparison of fine-tuning strategies across model scales (35M, 150M, 650M) and fusion methods (Token-wise vs. Channel-wise).

| **Fine-tune Strategy** | **Token-wise Concat** | | | **Channel-wise Concat** | | |
|---|---|---|---|---|---|---|
| (Tunable Parameters) | 35M | 150M | 650M | 35M | 150M | 650M |
| ESM2 | 0.333 | 0.4 | 0.425 | 0.333 | 0.4 | 0.425 |
| Full Fine-tune (100%) | 0.435 | 0.456 | 0.463 | 0.305 | 0.378 | 0.029 |
| LoRA + Adapter (5-10%) | 0.443 | 0.469 | 0.465 | 0.296 | 0.395 | 0.005 |
| Adapter-only (∼1%) | 0.358 | 0.407 | 0.45 | 0.311 | 0.385 | 0.435 |

As Table 4 shows, the first important message is that more tunable parameters *do not* correspond to better performance. Concretely, in the Token-wise Concat method, the LoRA + Adapter strategy performs best across all three scales of the pre-trained model, followed by Full Fine-tune and Adapter-only strategy. In the Channel-wise Concat method, however, excessive adjustable parameters can even severely impair performance, suggesting that over-tuning the model can destroy previously learned knowledge during pre-training. Secondly, the optimal fine-tune strategy is not generalizable and can be affected by the feature fusion method and even the model size. For example, the LoRA + Adapter strategy performs best in the Token-wise Concat method, while in the Channel-wise Concat method, the LoRA + Adapter strategy (0.296) is worse than the Full Fine-tune (0.305) and the Adapter-only (0.311) methods in the 35M backbone. In terms of the 150M backbone, the LoRA + Adapter strategy obtained the best performance (0.395) with respect to the Full Fine-tune (0.378) and Adapter-only (0.385) strategies, reflecting the high variance of the same fine-tuning strategy on different fusion methods and scales of the backbone model.

Third, clear scaling behavior, i.e., consistent performance improvements as the backbone grows, appears primarily under the Adapter-only setting. While prior work has reported benefits from increasing pretrained model scale for multimodal fine-tuning (Wang et al., 2024; Shukor et al., 2025), our experiments show that this monotonic scaling is mainly observed when only the adapters are tuned. The results of the Adapter-only strategy show a clear increase path as the model scales, both on Token-wise Concat and Channel-wise Concat. For LoRA+Adapter and Full Fine-tune, gains are often present for smaller backbones but can stagnate or become unstable on larger ones, suggesting potential overfitting or interference with pretrained weights. A plausible interpretation is: when the PLM backbone is small, extra adaptation capacity (LoRA updates or full tuning) is needed for the model to absorb structural signals; when the backbone is large, the pretrained model already has sufficient representational power to integrate structure, and minimal interventions (adapter-only) are both sufficient and more stable.

Table 5: Ablation results on the 35M backbone with LoRA + Adapter tuning. The top block varies the depth of the projection MLP used to map structural embeddings. The bottom block compares structure encoders.

| | | Average | Activity | Binding | Expression | Fitness | Stability |
|---|---|---|---|---|---|---|---|
| Layers | $2 \times$ MLP | 0.443 | 0.413 | 0.387 | 0.449 | 0.35 | 0.614 |
| | $3 \times$ MLP | 0.443 | 0.412 | 0.384 | 0.45 | 0.351 | 0.617 |
| | $4 \times$ MLP | 0.441 | 0.411 | 0.384 | 0.445 | 0.349 | 0.614 |
| Encoder | ProteinMPNN | 0.42 | 0.404 | 0.352 | 0.429 | 0.327 | 0.589 |
| | ESM-IF | 0.443 | 0.412 | 0.384 | 0.45 | 0.351 | 0.617 |
| | ProteinMPNN + ESM-IF | 0.437 | 0.408 | 0.379 | 0.445 | 0.343 | 0.609 |

### 4.4 ABLATION ON MAIN COMPONENTS

To understand how component design influences the performance, we conducted ablations on adapter layers and structure encoders. Table 5 reports controlled ablations on two axes: the depth of the MLP used to project structural features, and the choice/combination of structure encoders. All experiments are performed on the 35M backbone, and trained using the LoRA + Adapter strategy.

For the MLP depth, results are very close across 2, 3, and 4 layers. There is no clear improvement as we add depth: 2×MLP and 3×MLP give essentially the same average performance, while 4×MLP shows a tiny drop. Task-wise differences are also minor (3×MLP marginally helps expression and stability), but the overall gains are negligible compared with the extra parameters and computational cost. Practically, a shallow MLP is sufficient and more efficient; based on this, we use 3 layers as our default setting.

For the structure encoder, ESM-IF is the strongest single encoder in our setup, improving average performance and several tasks relative to ProteinMPNN. Interestingly, concatenating both encoders does not produce additive gains; the combined setup performs slightly worse than ESM-IF alone on most metrics. This suggests the encoders carry overlapping or even conflicting signals when naively merged; simple concatenation without per-encoder gating or attention can make it harder for the PLM to extract the most useful structural cues.

In summary, the ablations indicate that using a stronger, better-aligned structure encoder is a promising direction; we therefore leave more systematic exploration of encoder combinations and smarter fusion mechanisms (e.g., per-encoder gating or attention-based fusion) to future work.

## 5 DISCUSSION

In this paper, we show that multimodal fine-tuning of pretrained protein language models is a practical, effective way to bring structural information into PLMs. Specifically, we introduce InstructPLM-mu and evaluate three fusion designs, including Cross Attention, Channel-wise Concat, and Token-wise Concat. Our experiments demonstrate that multimodal fine-tuning consistently improves zero-shot mutation prediction over sequence-only baselines, and is competitive with several stronger multimodal methods trained from scratch. Notably, we find that the choice of fusion is critical: Token-wise Concat delivers the most robust gains across multiple backbone scales and different downstream functional categories. Further more, extensive ablations evidenced that the fine-tuning recipe is also crucial: parameter-efficient schemes (e.g., LoRA + adapters) often provide the best trade-off between performance and cost, whereas overly aggressive updates (full fine-tuning) can harm larger backbones and lead to catastrophic forgetting.

Despite these positive results, several limitations point to clear directions for future work. First, stronger or more diverse structural encoders and better encoder-level fusion (for example, per-encoder gating or attention) may unlock further gains beyond what simple concatenation provides; we leave systematic exploration of such fusion mechanisms to future work. Second, more fine-grained tuning protocols can be investigated. Such as staged training schedules, layer-wise unfreezing, or hybrid update schemes, can be performed to address the instability and catastrophic forgetting we observe when adapting very large backbones. We consider these directions important next steps to broaden and solidify the practical utility of multimodal fine-tuning for protein modeling.

## 6 REPRODUCIBILITY STATEMENT

All datasets used in this paper are publicly available, and preprocessing steps, including sequence filtering and train/validation/test splits, are described in Section 4.1. Implementation details of the InstructPLM-mu model, training hyperparameters, and evaluation protocols are provided in Section 3 and Appendix A.1. Complete results for all benchmarks, along with ablation studies, are reported in Appendix A.2. We provide an anonymized repository[1] containing the evaluation source code and instructions to reproduce the experiments.

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

# A APPENDIX

## A.1 TRAINING DETAILS

**Backbone model.** We adopt the publicly released ESM2 protein language models as our backbone. Three different parameter scales are explored: 35M[2], 150M[3], and 650M.[4] All checkpoints are initialized from the official HuggingFace releases without additional pre-training.

**Structure encoder.** We adopt the publicly released ProteinMPNN[5] and ESM-IF1[6] (ESM-Inverse Folding) checkpoints as our structure encoders. Specifically, we concat 4 vanilla models of ProteinMPNN `v_48_002.pt`, `v_48_010.pt`, `v_48_020.pt`, `v_48_030.pt`. Input 3D coordinates are preprocessed following the official ProteinMPNN pipeline (atom type filtering and coordinate centering).

**Multi-model Projector.** To integrate sequence and structural representations, we employ a lightweight multi-modal projector implemented as a multi-layer perceptron (MLP) with GELU (Hendrycks & Gimpel, 2016) activation. The projector maps the concatenated embeddings from the protein language backbone and the structure encoder into a unified latent space with a dimension of the backbone model. Layer normalization is applied after each hidden layer.

**Low-Rank Adapter.** For efficient fine-tuning of the backbone model, we insert Low-Rank Adaptation (LoRA) modules into every linear layer of the transformer blocks. The LoRA rank is set to 32 and the scaling factor ($\alpha$) to 256. The adapters are trained jointly with the projector while all original backbone weights remain frozen.

Table 6: Summary of training hyperparameters for InstructPLM-mu.

| Hyperparameter | Value |
|---|---|
| Learning rate | 1e-4 |
| Batch size | 256 |
| Number of training epochs | 20 |
| Optimizer | Adam |
| Warm-up steps | 100 |
| Weight decay | 1e-1 |

**Other Hyperparameters.** Key hyperparameters used in training are summarized in Table 6.

## A.2 EVALUATION AND MORE RESULTS

**Metrics.** Model performance is evaluated using the Spearman rank correlation coefficient (Spearman's $\rho$) between the predicted mutation effects and the experimentally measured ground-truth scores. Spearman's $\rho$ measures the monotonic relationship between two variables and is insensitive to the absolute scale of the predictions, making it well-suited for assessing whether the model correctly ranks protein variants by functional effect rather than merely matching their exact values.

**Results per assay.** For each assay, we report the Spearman correlation across all tested variants. Figures 4, 5, and 6 show the performance of the baseline models (ESM2 (650M) and ESM3) as well as our fine-tuned methods. For clarity, we highlight the results of the Token-wise Concat model with the ESMif encoder and trained with the LoRA + adapter strategy. The figures indicate that the improvements are most pronounced on proteins for which the baseline model (ESM2) previously performed poorly (Figure 4), while still maintaining relatively high correlation on assays that were easier for the baseline (Figure 6).

---

[2] https://huggingface.co/facebook/esm2_t12_35M_UR50D
[3] https://huggingface.co/facebook/esm2_t30_150M_UR50D
[4] https://huggingface.co/facebook/esm2_t33_650M_UR50D
[5] https://github.com/dauparas/ProteinMPNN
[6] https://huggingface.co/facebook/esm_if1_gvp4_t16_142M_UR50

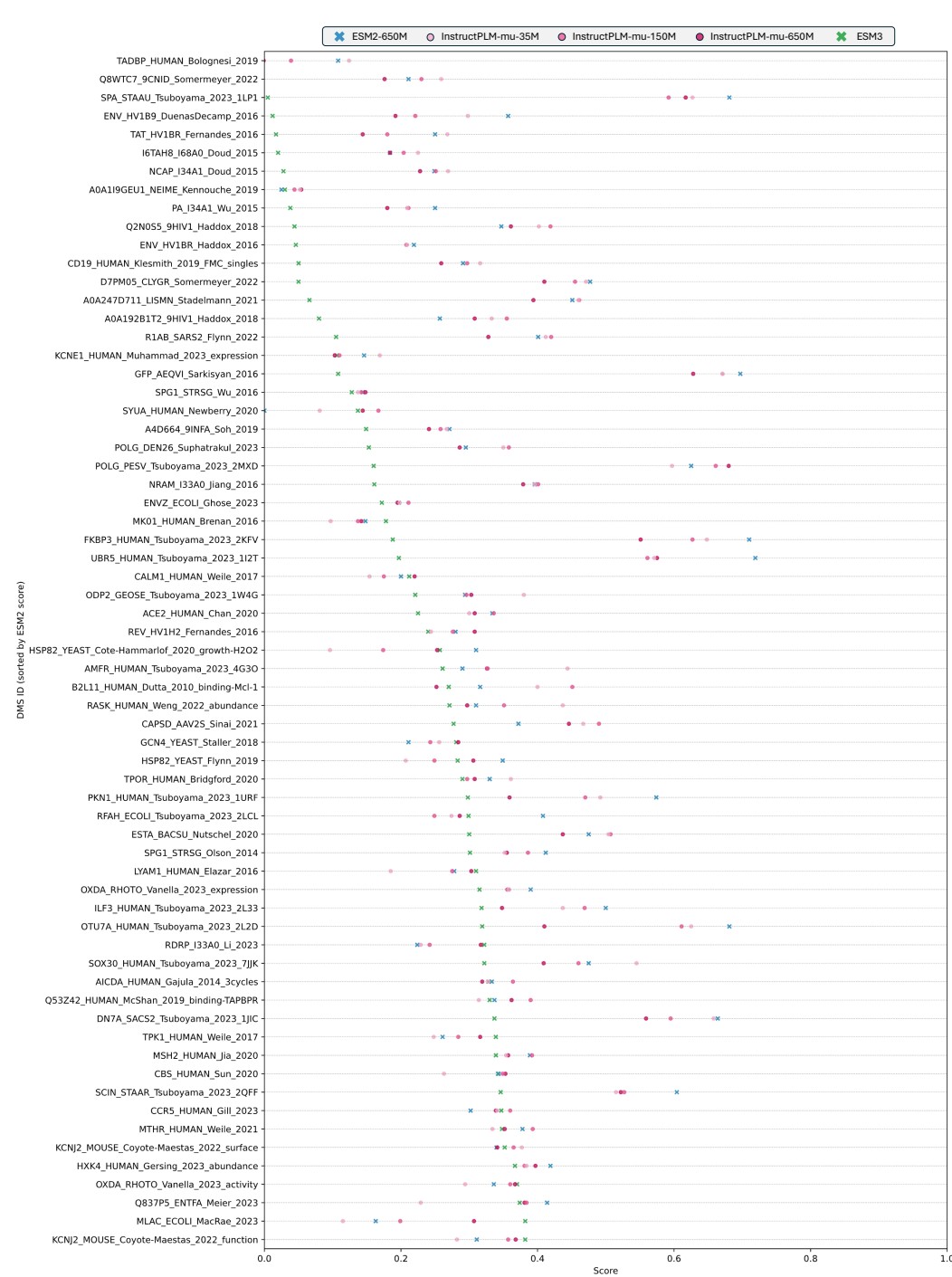

Figure 4: Spearman correlations on individual DMS datasets, sorted by ESM2 (650 M) performance. Baselines use cross markers; InstructPLM-mu are shown as colored circles.

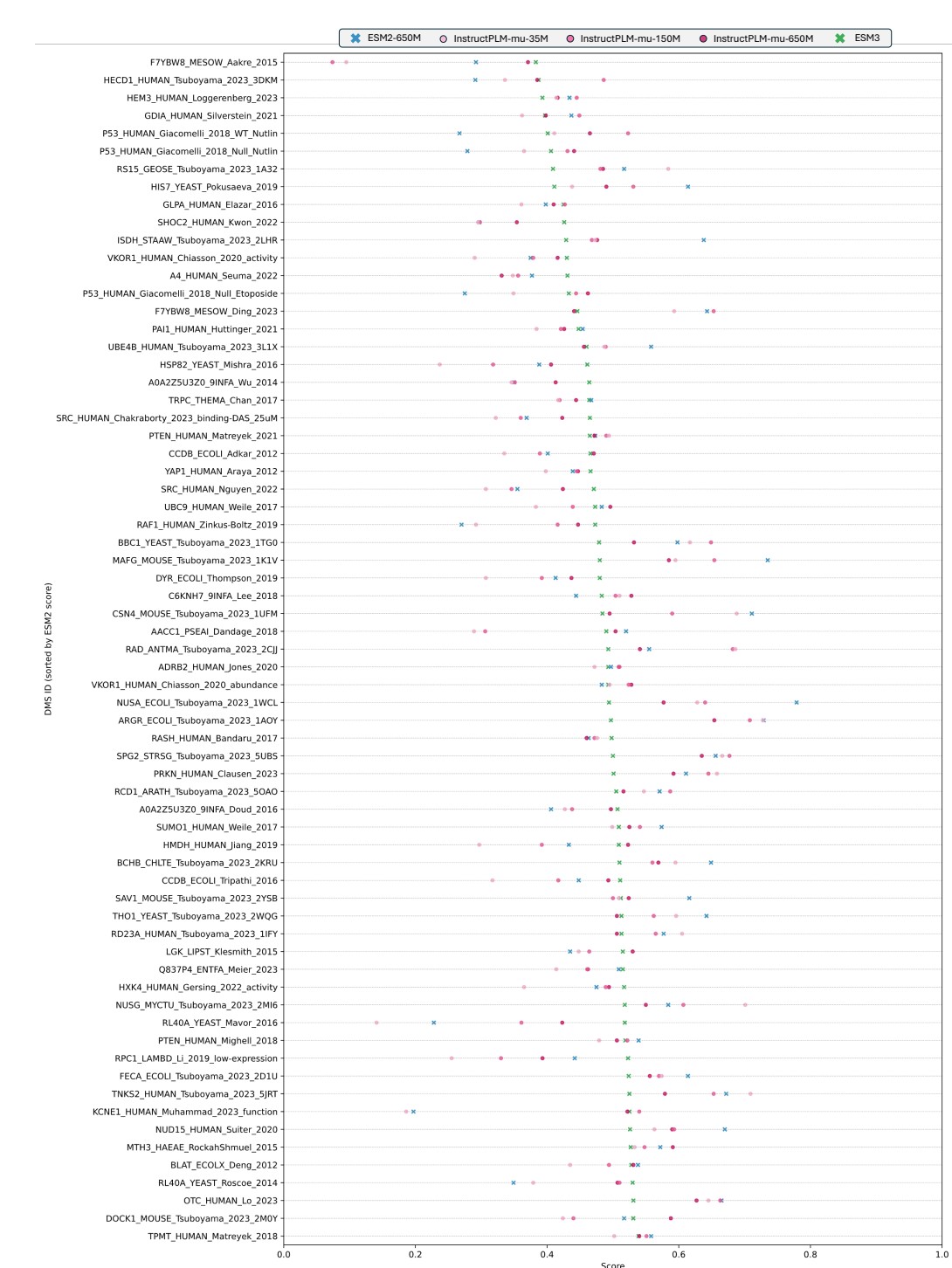

Figure 5: Spearman correlations on individual DMS datasets, sorted by ESM2 (650 M) performance. Baselines use cross markers; InstructPLM-mu are shown as colored circles.

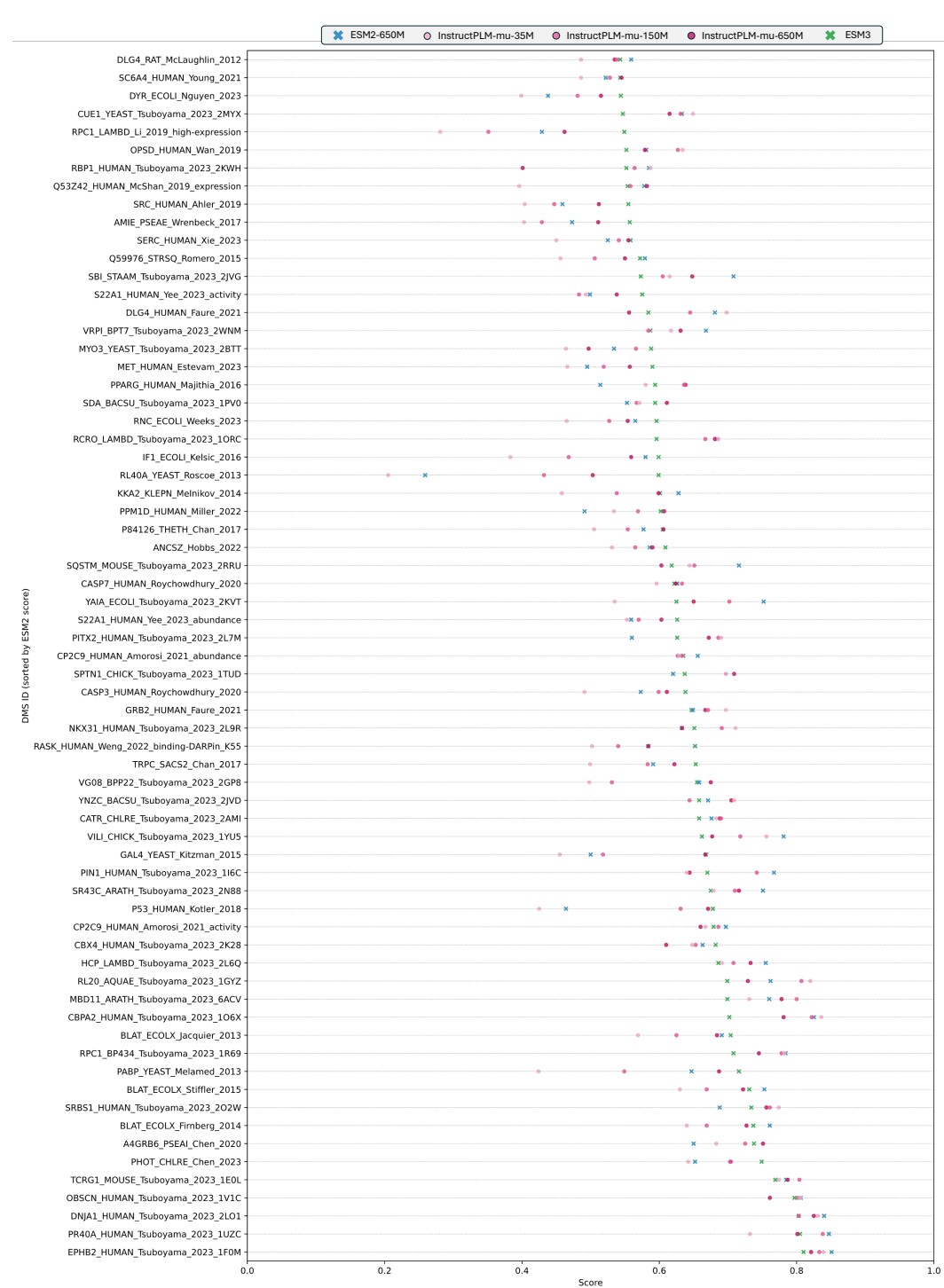

Figure 6: Spearman correlations on individual DMS datasets, sorted by ESM2 (650 M) performance. Baselines use cross markers; InstructPLM-mu are shown as colored circles.

