# OpenReview forum: "InstructPLM-mu: 1-Hour Fine-Tuning of ESM2 Beats ESM3 in Protein Mutation Predictions"
_ICLR.cc/2026/Conference — ICLR 2026 Conference Withdrawn Submission_

### Official Review · Reviewer_aS8h · 2025-10-20

**Soundness:** 2
**Presentation:** 2
**Contribution:** 1
**Rating:** 2
**Confidence:** 4

**Summary:**

To understand the function of proteins, biologists perform deep mutational scans (DMSs). A flurry of work over the last two decades has shown that machine learning models trained on evolutionary sequence data can predict the outcome of DMS experiments from sequence alone. Recent work has shown that by introducing protein structure predictions can be further improved, leading to state-of-the-art DMS predictions.

This work considers taking pre-trained sequence-only models and incorporating structure information by fine-tuning. they consider 3 off-the-shelf fusion methods for incorporating structure information into these models. Their main result is that with trivial compute, one can improve the predictions of ESM2 (a sequence-only model) to make them competitive with ESM3 (a multi-model sequence-structure model). In tables 3 and 4 they also do some ablations, showing

**Strengths:**

* The figures clearly show their experimental setups
* With little compute, their models show non-trivial improvements in DMS prediction.

**Weaknesses:**

The main weakness of the paper is that it is not clearly or soundly argued what the significance of this result is.
 * On one hand, the authors pit "end-to-end" training against fine-tuning, and claim that their results support the later as a paradigm. Yet they only perform experiments on the ESM2 backbone.
* On the other, their fine-tuned model cannot stand on its own -- there remains a large gap between their model and other methods as shown in table 3.

There are other obvious experiments missing if one were suggesting that a practitioner should go out and fine-tune their method to automatically get better Spearmans.
* It's impressive the benefit they get after one hour but does this saturate at this time? What about different training times?
* Tables 4 and 5 show to me that the results are sensitive to fusion choices

I do not want the authors necessarily do all the experiments I suggest above.
Instead I would like the authors to first clearly lay out how they imagine a practitioner making use of the knowledge gained in their experiments.
Then I would like them to fill in the logical gaps specific to their argument with new experiments.

**Questions:**

In addition to the questions above, please consider:
* Given ProSST is already a strong and state-of-the-art multi-modal model, why should we bother fine-tuning sequence-only models? Maybe in the future all models will be pre-trained on multi-modal data.
* If you're suggesting that future methods should be pre-trained on sequence-only data then fine-tuned on fusion data (are you suggesting this?) then how do your results back this up? Can you quantify the overhead of pre-training on fusion vs sequence-only data?

---

### Official Review · Reviewer_P6b8 · 2025-10-27

**Soundness:** 3
**Presentation:** 2
**Contribution:** 2
**Rating:** 4
**Confidence:** 3

**Summary:**

This paper presents InstructPLM-mu, a multimodal fine-tuning framework for injecting structural information into pretrained protein language models (PLMs).
Instead of training large multimodal models such as ESM3 from scratch, the authors explore whether a lightweight fine-tuning of the sequence-only ESM2 can achieve comparable performance on mutation-effect prediction tasks.
The study provides practical guidance for efficient multimodal adaptation of PLMs and an interesting empirical observation that over-tuning can cause catastrophic forgetting.

**Strengths:**

Practical significance: Shows that multimodal fine-tuning of sequence-only PLMs can rival state-of-the-art models trained from scratch, greatly lowering computational barriers.

Systematic methodology: Compares multiple fusion and fine-tuning designs under controlled settings; results are reproducible and well-documented.

Clear empirical findings: Identifies the Token-wise Concat + LoRA Adapter combination as the best trade-off and reveals non-monotonic behavior between tunable-parameter count and performance.

Strong presentation: Figures, quantitative tables, and ablation studies are comprehensive and easy to follow.

Community value: Offers an efficient recipe likely to be adopted by researchers with limited compute in structural biology and multimodal modeling.

**Weaknesses:**

Limited and incremental novelty:
The main limitation of this work lies in its lack of conceptual novelty. The challenge of integrating structural information into sequence-based protein language models has already been widely studied, and several prior works have effectively addressed this issue through various multimodal alignment or fine-tuning strategies.
Compared with these efforts, the present paper does not introduce a fundamentally new modeling paradigm or learning objective. Its contribution is primarily engineering-oriented, focusing on a systematic comparison and empirical verification of existing fusion and fine-tuning techniques. While these comparisons are useful, they do not substantially advance the conceptual state of the field.

Lack of mechanistic insight:
The study convincingly demonstrates that the Token-wise Concat strategy performs best, yet provides limited explanation of why this happens. Deeper analyses, such as examining attention distributions, structural-token utilization, or representational alignment between sequence and structure, would significantly strengthen the paper’s interpretability and scientific insight.

Narrow evaluation scope:
The experimental validation is restricted to the ProteinGym benchmark. Although this is a standard dataset, it mainly covers mutation-effect prediction and does not fully reflect broader applications such as protein design, stability prediction, or binding-affinity estimation. Additional results on these downstream tasks would improve the generalizability and impact of the conclusions.

Minor reproducibility and reporting issues:
The anonymized code link prevents immediate verification; clearer runtime statistics, memory cost, and scaling behavior across backbone sizes would help readers better assess the claimed one-hour fine-tuning efficiency.

**Questions:**

Could the authors provide qualitative or visualization analyses (e.g., attention heatmaps) to explain how structure tokens influence sequence representations?

How sensitive is performance to the structure encoder choice (ProteinMPNN vs. ESM-IF) when scaling to larger backbones?

Would staged fine-tuning (e.g., layer-wise unfreezing) further mitigate catastrophic forgetting?

Could this instruction-style fine-tuning extend beyond structure—e.g., incorporating experimental assay metadata or evolutionary profiles?

Are there plans to release a small, ready-to-use toolkit for community replication?

---

### Official Review · Reviewer_deiR · 2025-10-29

**Soundness:** 2
**Presentation:** 3
**Contribution:** 1
**Rating:** 2
**Confidence:** 4

**Summary:**

The paper explores the question of “Can structure+sequence multimodal fine-tuning to a pretrained protein language model achieve comparable performance as multimodal model trained from scratch?” Authors fine-tune ESM2 on structure embeddings of ProteinMPNN or ESM-IF. Authors compare 3 different structure + sequence embedding fusion strategies: cross attention, channel-wise concat, token-wise concat. Training target is MLM objective. Training/validation is done on CATH4.3, evaluation is done one ProteinGym (zero-shot mutation effect prediction).
Through experiments on multiple scales of ESM2, the paper argues token-wise concat delivers the most gains.

**Strengths:**

Authors conduct a systematic comparison of fine-tuning strategies (both fusion methods and LoRA) across different model scales. This is valuable, as such comparisons are often done inconsistently or arbitrarily in previous works.

**Weaknesses:**

1. The overall research question and evaluation setup are not particularly compelling. Fine-tuning is typically meant to instill the model with new capabilities. The paper would be more appealing if it explored how structural fine-tuning could introduce new abilities or insights into large pLMs, rather than merely matching ESM3’s performance in one task. And that task is something model can already do (zero shot fitness prediction).
2. Three fusion methods explored are quite standard and straightforward. The paper mainly compares three fusion strategies and reports which performs best, without offering deeper insights.
3. Even the empirical analysis is somewhat limited, since the experiments are only conducted on ESM2. Broader validation (e.g., across models or datasets) would strengthen the claims.

**Questions:**

1. If we account for the compute used to train both the structure encoder and the language model separately, is this approach actually more efficient than training a single multimodal model end-to-end?
2. Why not explore the opposite? What happens if Protein-MPNN or ESM-IF is fine-tuned to fuse ESM2 embedding?

---

### Official Review · Reviewer_EoJD · 2025-11-01

**Soundness:** 3
**Presentation:** 3
**Contribution:** 3
**Rating:** 6
**Confidence:** 4

**Summary:**

This paper presents InstructPLM-mu, a multimodal fine-tuning framework that efficiently injects structural information into pretrained sequence-only protein language models. Using ESM2 as the base, the authors evaluate three fusion strategies—Cross Attention, Channel-wise Concat, and Token-wise Concat—and show that Token-wise Concat achieves the best performance. After only one hour of fine-tuning, InstructPLM-mu on ESM2 (150M) surpasses ESM3 in zero-shot protein mutation prediction. Ablation studies demonstrate that fine-tuning strategy and fusion design critically affect outcomes, with LoRA + Adapter achieving the best balance between cost and performance. The work highlights multimodal fine-tuning as a practical and efficient alternative to training large multimodal models from scratch.

**Strengths:**

**Strengths**

1. **Efficiency and performance:** Matches or exceeds ESM3 with one-hour fine-tuning on modest compute.
2. **Comprehensive analysis:** Systematic comparison of fusion and tuning strategies clarifies their distinct effects.

**Weaknesses:**

1. The work is solid and thorough but offers limited novelty, as it primarily adapts existing multimodal fusion strategies from other domains rather than introducing fundamentally new modeling concepts for protein language models.
2. Although the method yields clear gains over ESM baselines, it still falls short of the strongest structure-aware protein language models in overall performance.

**Questions:**

1. The random split of the CATH dataset is not sufficiently rigorous. In multimodal protein modeling, it is generally advisable to ensure splits respect both sequence identity and structural homology to prevent data leakage.
2. It would strengthen the evaluation if the authors included additional benchmarks, such as those used in SaProt (e.g., Table 2 benchmark) and ClinVar, to better demonstrate performance.
3. It is unclear whether the ProteinGym benchmark originally excluded long protein assays for ESM2. The authors should clarify this and ensure consistency with the original evaluation protocol.

---

### Note · Authors · 2026-01-15

I have read and agree with the venue's withdrawal policy on behalf of myself and my co-authors.